# Effect of Femtosecond Laser-Irradiated Titanium Plates on Enhanced Antibacterial Activity and Preservation of Bacteriophage Stability

**DOI:** 10.3390/nano13142032

**Published:** 2023-07-09

**Authors:** Liga Grase, Pavels Onufrijevs, Dace Rezevska, Karlis Racenis, Ingus Skadins, Jonas Karosas, Paulius Gecys, Mairis Iesalnieks, Arturs Pludons, Juta Kroica, Gediminas Raciukaitis

**Affiliations:** 1Institute of Materials and Surface Engineering, Faculty of Materials Science and Applied Chemistry, Riga Technical University, 7 Paula Valdena Street, LV-1048 Riga, Latvia; mairis.iesalnieks@rtu.lv (M.I.); arturs.pludons@rtu.lv (A.P.); 2Institute of Technical Physics, Faculty of Materials Science and Applied Chemistry, Riga Technical University, 7 Paula Valdena Street, LV-1048 Riga, Latvia; onufrijevs@latnet.lv; 3Department of Biology and Microbiology, Riga Stradins University, 16 Dzirciema Street, LV-1007 Riga, Latvia; dace.rezevska@rsu.lv (D.R.); karlis.racenis@rsu.lv (K.R.); ingus.skadins@rsu.lv (I.S.); juta.kroica@rsu.lv (J.K.); 4Department of Laser Technologies, Center for Physical Sciences and Technology, Savanoriu Ave. 231, LT-02300 Vilnius, Lithuania; jonas.karosas@ftmc.lt (J.K.); paulius.gecys@ftmc.lt (P.G.); g.raciukaitis@ftmc.lt (G.R.)

**Keywords:** Titanium, fs laser, LIPSS, oxide, antibacterial, bacteriophages

## Abstract

Titanium (Ti) is widely recognized for its exceptional properties and compatibility with medical applications. In our study, we successfully formed laser-induced periodic surface structures (LIPSS) on Ti plates with a periodicity of 520–740 nm and a height range of 150–250 nm. To investigate the morphology and chemical composition of these surfaces, we employed various techniques, including field emission scanning electron microscopy, energy dispersive X-ray spectroscopy, atomic force microscopy, X-ray photoelectron spectroscopy, and Raman spectroscopy. Additionally, we utilized a drop-shape analyzer to determine the wetting properties of the surfaces. To evaluate the antibacterial activity, we followed the ISO 22196:2011 standard, utilizing reference bacterial cultures of Gram-positive *Staphylococcus aureus* (ATCC 25923) and Gram-negative *Escherichia coli* (ATCC 25922). The results revealed enhanced antibacterial properties against *Staphylococcus aureus* by more than 99% and *Escherichia coli* by more than 80% in comparison with non-irradiated Ti. Furthermore, we conducted experiments using the *Escherichia coli* bacteriophage T4 (ATCC 11303-B4) and the bacterial host *Escherichia coli* (ATCC 11303) to investigate the impact of Ti plates on the stability of the bacteriophage. Overall, our findings highlight the potential of LIPSS on Ti plates for achieving enhanced antibacterial activity against common bacterial strains while maintaining the stability of bacteriophages.

## 1. Introduction

Lasers play a crucial role as versatile tools in the surface modification of various materials, mainly at the macroscale and microscale. However, laser processing at the nanometer scale is developing very rapidly [1]. Some examples of nanoscale laser processing include laser nanolithography [2], laser ablation for nanoparticle synthesis [3], and laser-induced surface modifications [4,5,6]. Among these techniques, laser-induced surface modifications offer precise control over surface relief and phase composition, thereby influencing a wide range of properties such as wettability, antimicrobial characteristics, and tribology. One technique used for surface patterning is the creation of laser-induced periodic surface structures (LIPSS), also known as ripples. This method involves the use of a laser beam with linearly polarized radiation [7,8,9]. LIPSS can be classified based on their spatial frequency into two types: low spatial frequency LIPSS (LSFL) and high spatial frequency LIPSS (HSFL). In LSFL, the period Λ of the structures is approximately equal to the laser wavelength (Λ~λ), and their orientation is perpendicular to the polarization of the laser pulse [10]. In HSFL, the period Λ is much smaller than the laser wavelength (Λ << λ), and the structures are generated with an orientation parallel to the polarization of the laser light [11].

The exact mechanism behind LIPSS formation is still a topic of debate and can generally be categorized into two types of theories based on the materials and laser parameters used: electromagnetic theories and matter reorganization theories [4,12]. Electromagnetic theories describe the deposition of optical energy into the solid, while matter reorganization theories are based on the redistribution of matter in the surface layer. One of the most widely accepted mechanisms for the formation of LSFL structures involves the interaction of surface plasmon polaritons with rough metal surfaces [13]. The initial roughness of the material plays a crucial role in producing scattering, which can generate surface plasmon polaritons that interfere with the incident light. This interference modulates the absorbed fluence and selectively removes material to create parallel periodic structures [7].

The LIPSS technique is applicable to a wide range of materials, including metals [14], semiconductors [15], superconductors [16], polymers [17,18], dielectrics [19], and 2D nanomaterials [20]. Surfaces of metals created using LIPSS hold significant potential for medical applications, including antimicrobial, self-cleaning, antifriction, and antifogging properties. Femtosecond lasers can also be utilized to produce ultrasensitive surface-enhanced Raman spectroscopy (SERS) platforms based on silicon for biomedical applications [21]. 

In particular, the antimicrobial properties have garnered significant attention, especially considering the COVID-19 pandemic, as they serve as a catalyst for heightened awareness and future concerns regarding potential outbreaks. The controlled enhancement of antibacterial properties of biomedical materials can be achieved by using laser-induced micro-texturing of the surfaces [5]. LIPSS play an important role in the production of antibacterial surfaces [22]. Various surface characteristics, such as wettability, roughness, topography, surface charge, and stiffness, can influence the adhesion of bacteria [23]. However, the development of the antibacterial properties of a surface depends not only on the parameters of the obtained microstructure but also on certain types of bacteria. For example, Gram-positive and Gram-negative bacteria can be classified based on their cell wall structure [24]. Gram-positive bacteria have a thick peptidoglycan layer in their cell wall, while Gram-negative bacteria have a thinner peptidoglycan layer and an outer membrane containing lipopolysaccharides. Overall, various antimicrobial mechanisms can be identified, such as physical antiadhesion and chemical elimination [25]. 

Titanium is a well-known material for medical applications due to its superior properties and biocompatibility with living cells. However, their limitations lie in their inability to stimulate new bone formation and their lack of antibacterial properties [26]. To address the latter, researchers have been focusing on surface modifications of Ti, aiming to transform its surface layer into an oxide or apply alternative films to obtain antibacterial coatings. These modifications are pursued to enhance the functionality of Ti implants and mitigate the risk of implant-related infections. It is also well known that after irradiation with a laser, it is possible to form a layer of TiO_2_ with a controllable phase composition [27] on a titanium surface. Such a layer could also lead to enhanced antimicrobial activity due to the presence of radicals and superoxides, which could contribute to the damage of the outer cells of the microorganisms. In addition, research can be found regarding multi-phase titanium oxide LIPSS formation under fs laser irradiation on thin titanium films [28].

Regardless of the extensive use of titanium and its alloys for biomedical implants, the surfaces of implants may serve as a critical determinant for biofilm formation and subsequent implant-associated infections [29,30]. Furthermore, bacterial biofilms and the overuse of antibiotics contribute to the development of antibiotic resistance [31]. Hence, it is urgent to look for novel and alternative approaches to prevent, combat, and overcome the advancement of implant-associated infections [32]. One of the potential strategies might be antibacterial bacteriophage-based coatings on the surface of implants [33]. Bacteriophages (or phages) are prokaryote-specific viruses. Lytic phages are recognized as promising nontraditional antibacterials due to their biological characteristics, including the antibiofilm effect. To impregnate phages in coatings, it is crucial to maintain their stability when exposed to biomedical implant surfaces.

The study aims to develop femtosecond laser technology for obtaining enhanced antibacterial properties of titanium surfaces with laser-induced periodic surface structures against both Gram-positive *Staphylococcus aureus* and Gram-negative *Escherichia coli*, as well as preserving bacteriophage stability.

## 2. Materials and Methods

### 2.1. Materials and Laser Setup

Commercial titanium (Ti) samples, specifically grade 1 ASTM B265 with a purity of 99.6% or higher, were procured from Goodfellow Cambridge Ltd. [34]. The samples had dimensions of 25 × 25 × 2 mm. 

The samples were irradiated with a pulsed Yb:KGW femtosecond (fs) laser (model: Pharos-6W, Light Conversion, Vilnius, Lithuania) with the following parameters: wavelength 1030 nm, pulse duration 300 fs, max. pulse intensity 0.8 TW/cm^2^ (the optimal laser intensity for the formation of LIPSS), max. laser fluence 0.25 J/cm^2^, repetition rate of 600 kHz, beam diameter 40 µm, beam profile “Gaussian.” Linear laser beam polarization was used in all experiments. The laser power in the experiments was varied with an attenuator consisting of a half-wave plate and a polarizer. The scanning of the laser light was performed normally to the Ti surface at a speed of 1 mm/s. The irradiation of the samples was carried out at room temperature under ambient pressure. To investigate the impact on antibacterial properties, the study considered varying hatch spacings within the micrometer range, specifically 4, 8, 12, and 16 µm. Additional laser beam defocus of 2 mm was implemented to enlarge the spot size up to 80 µm. The experimental setup for this process is illustrated in Figure 1. 

### 2.2. Characterization of Titanium Plates before and after Irradiation

#### 2.2.1. Scanning Electron Microscopy (SEM) Analysis

A field emission scanning electron microscope (FESEM) FEI Nova NanoSEM650 (Eindhoven, The Netherlands) equipped with an energy dispersive X-ray (EDX) analysis detector EDAX^TM^ (Pleasanton, CA, USA) was used to characterize surface structures and perform chemical analysis. To prevent the buildup of charge on the surface of the samples without the need for a conductive coating, a low vacuum mode 60 Pa and Helix^TM^ (Singapore) detector was used. 

#### 2.2.2. X-ray Photoelectron Spectroscopy (XPS) Analysis

The effect of surface modifications was analyzed using X-ray photoelectron spectroscopy (XPS, Escalab Xi+, Thermo Scientific, Waltham, MA, USA) with an Al K-alpha X-ray source without further surface cleaning. XPS spectra were collected before and after surface etching. Surface etching was performed using an Ar ion gun with mild sputtering conditions for 10 s. The amount of surface contamination decreases with Ar ion sputtering, as is expected. The advantageous carbon peak for C-C at 284.8 eV was used as a calibration point. Peak fitting was performed using Avantage 5.9925 software.

#### 2.2.3. Raman Spectroscopy

The Raman shift measurement was performed at room temperature using the Renishaw In-ViaV727 spectrometer in a backscattering configuration. The phonon excitation was induced using a green laser (Ar+, λ = 514.5 nm) with a grating of 1200 mm^−1^. Each spectrum was recorded from three accumulations, with an exposure time of 3 s for each accumulation.

#### 2.2.4. Atomic Force Microscopy (AFM)

Atomic force microscopy (AFM) was used to characterize LIPSS structures and examine the overall roughness of the titanium surfaces. The Vecco CP-II AFM equipment was used for the examination of topography. The IP21 software was used for both data analysis and calculations of the LIPSS periods. 

### 2.3. Wettability

A drop-shape analyzer was used to determine wetting properties. The contact angle measurements were performed using the KRÜSS Drop Shape Analyzer DSA25E. Five parallel measurements were conducted at room temperature in the air. Contact angle measurements were carried out using a water droplet substance. The volume of the liquid drop was fixed at 10 microliters for deionized water. The contact angle of the surface of the sample was analyzed as received just after irradiation. 

### 2.4. Antibacterial Properties

The antibacterial activity of samples was determined by the ISO 22196:2011 standard using reference bacterial cultures of Gram-positive *Staphylococcus aureus* (ATCC 25923) and Gram-negative *Escherichia coli* (ATCC 25922) commercially obtained from The American Type Culture Collection (ATCC). 

### 2.5. Preparation of Bacteriophage Solution and Its Exposure to Titanium Plates

The *Escherichia coli* bacteriophage T4 (ATCC 11303-B4) and its bacterial host, *Escherichia coli* (ATCC 11303), were used to evaluate the effect of Tref and T16 titanium plates on the stability of the bacteriophage over a 36 h period.

#### 2.5.1. Preparation of the Bacteriophage Solution

The reference phage stock was recovered from a frozen vial in the presence of the recommended and respective bacterial broth cultures, following the general procedures provided by the ATCC. The recovered phage stock was propagated twice to attain a higher viral titer. Briefly, the propagation procedure included a set of the following steps: (a) flood of the webbed plates with trypticase soy broth (TSB) after the initial plaque assay; (b) chloroform (CHCl_3_) treatment to remove the soft agar overlay and collected supernatant; (c) refrigeration; (d) centrifugation; and (e) 0.20 µm filtration to obtain phage lysate. The viral titer was estimated using a plaque assay. The procedure involved serial 10-fold dilutions of phage stock and plating with a respective bacterial host by applying a soft-agar overlay. Individual plaques, namely viable phage particles, observed on overnight incubated plates were counted, and the viral titer was expressed in plaque-forming units per milliliter (PFU/mL).

#### 2.5.2. Preparation of Mixtures of Bacteriophage Solutions and Titanium Plates

The phage lysate was diluted with TSB to acquire an increased volume of phage stock. Within 5 mL of freshly acquired phage suspension, Tref and T16 titanium plates were immersed. A control with 5 mL of phage-only suspension was provided. Plastic centrifuge tubes with prepared experiment samples were incubated at 37 °C with shaking at 150 rpm for various time periods—0 h, 6 h, 12 h, 24 h, and 36 h, respectively.

#### 2.5.3. Determination of Bacteriophage Stability

The stability performance of the phage in the presence of Tref and T16 titanium plates was evaluated by executing the plaque assay at the aforementioned multiple incubation points in time—0 h, 6 h, 12 h, 24 h, and 36 h, respectively.

#### 2.5.4. Statistical Analysis

Duplicate experiments were executed to enhance accuracy. The Mann–Whitney U test ensured that prepared mixtures (phage-only suspension in TSB, phage suspension with Tref, and phage suspension with T16) were compared. The results were expressed in the form of mean ± standard deviation. Statistical analysis was performed using GraphPad Prism software, version 9. The tests resulted in *p*-values greater than 0.05, considering the differences in the results obtained as not statistically significant.

## 3. Results and Discussion

Bacterial adhesion to a material surface is fundamentally essential for surface contamination to occur. Adhesion depends both on bacterial factors and surface properties. By altering the surface properties, the adhesion of bacteria to the surface can be effectively prevented. The adhesion of bacteria is significantly influenced by surface morphology. Nano- and micro-structures significantly increase the contact adhesion area, resulting in more effective bactericidal properties compared to flat surfaces [35]. Morphological differences observed in FESEM images (Figure 2a–d) show that after irradiation with a femtosecond laser, the LIPSS were formed on the surface of titanium plates. The laser parameters are shown in Table 1. The periodicity (Λ) of these structures was determined using the two-dimensional (2D) Fast Fourier Transform (FFT) method, assisted by the open-source software Gwyddion, Version 2.63 (Figure 2e–h). To analyze the periodicity of LIPSS, the 2D-FFT profiles were examined (Figure 2i–l). This was achieved by measuring the distance between the two most intensive mirror-like related peaks, which were fitted using the Lorentz function. The formula used to calculate Λ is Λ = 2/(f_1_ − f_2_), where f_1_ and f_2_ represent the spatial frequencies observed in the 2D-FFT profile. Periodicity of LIPSS, which was in the range of about 520–740 nm, depending on hatch spacing used (Λ_T4_—538 ± 30 nm; Λ_T8_—520 ± 51 nm; Λ_T12_—660 ± 17 nm; Λ_T16_—740 ± 24 nm). Such structures can be classified as low spatial frequency LIPSS (LSFL). The LSFL period depends on many parameters like laser fluence, accumulated irradiation dose, hatch distance, and pulse overlap. The range of 520–740 nm is typical for LIPSS formed at 1030 nm wavelength [36]. Such structures can also be obtained by combining LIPSS with the direct laser interference patterning (DLIP) method [37] in order to obtain multilevel hierarchical surface structures. Additionally, nanocrystals with random orientation and morphology can be observed on the surface of the structures.

Based on the 2D-FFT images presented in Figure 2, it is evident that the LIPSS exhibited a consistent orientation, indicating that the LIPSS were perpendicular to the polarization direction. These findings align with previous observations reported in the literature, where it is commonly noted that the LSFL are oriented perpendicular to the direction of laser polarization. In contrast, the HSFL align parallel to the direction of laser polarization [7].

EDS mapping (Figure 3) shows that after irradiation with a fs laser, the oxygen concentration on the surface of titanium increases in comparison with nonirradiated samples, where traces of native oxide [38] can be observed (Figure 3a). It can also be observed that oxygen is mainly distributed on the top surface of LIPSS. 

The AFM study of the samples allows us to determine more accurately the quantitative measures of LIPSS shape and surface roughness before studying antibacterial properties. AFM scans are presented in Figure 4 for laser-irradiated samples with LIPSS. It can be noted that by irradiating the sample with a Yb:KGW fs laser, the LIPSS—periodically arranged hills and valleys with the periodicity around Λ = 520–740 nm—are formed on the surface. The average value of the peak-to-valley height difference is 150–250 nm. A slight surface polishing has been achieved due to the reduced roughness root mean square R_rms_ value from 130 nm to 50–70 nm for the non-irradiated and irradiated samples, depending on hatch spacing. It is presumed that the presence of numerous sharp structures obtained by fs laser on the rough surface is likely to be destructive to bacteria [39]. Furthermore, the conical shape of LIPSS results in a smaller adhesion area for bacteria to attach to in comparison with flat surfaces.

We employed reference data on their frequencies in single crystals to identify titanium oxide-related modes on Raman spectra after laser processing [40]. Nevertheless, analyzing the Raman spectra of amorphous or polycrystalline layers can be challenging due to peak shifting and broadening compared to those observed in single crystals. These alterations in peak characteristics are primarily attributed to the presence of grain boundaries, extended defects, and stresses. The Raman spectra (Figure 5) of samples T4, T8, T12, and T16 with different hatch spacing show amorphous semiconductor phase formation with a broad peak around 250 cm^−1^ [41]. Alternately, it could be attributed to the formation of nanocrystallites on the surface of LIPSS. Additionally, in two cases, the influence of the utilization of variant laser parameters in the case of a hatch spacing of 12 µm was investigated. For this purpose, we used a higher laser fluence of 106 mJ/cm^2^ for T12WL samples (where WL stands for without LIPSS) and 254 mg./cm^2^ for T12A samples (where A stands for ablation). While T12WL remains in a transitional state between the amorphous and crystalline phases, the sample surface of T12A corresponds to the formation of the TiO_2_ rutile phase [40].

From FESEM images (Figure 6), it can be seen that sample T12WL can be characterized by nonregular surface structures, while in the case of T12A, ablation of TiO_2_ nanoparticles (rutile phase) takes place. The morphology of nanoparticles exhibits a diverse size distribution, with the nanoparticles typically being less than 70 nm in diameter, while Figure 6b inset provides visual evidence that these nanoparticles tend to have a round shape. The antimicrobial properties of such structures were not tested due to the absence of LIPSS in the case of T12WL, as well as the formation of nanoparticles via the ablation process in the case of T12A, which has weak adhesion to the titanium plate surface. The size distribution observed in the morphology of nanoparticles makes it an intriguing case for developing a method for metallic or semiconductor nanoparticle production with various sizes from metals using laser [3].

For chemical bond detection and surface atomic concentration analysis, XPS high-resolution core level spectra were scanned for Ti and O in non-irradiated and irradiated areas (Figure 7). The O1s peak at 531 eV can be attributed to OH group formation or defective oxides [42]. After surface etching using an Ar ion gun with mild etching conditions, lesser titanium oxides and titanium metal can be detected. Peak splitting between Ti (IV) 2p3/2 and 2p1/2 peaks cannot be attributed to rutile or anatase phase due to low signal strength. The peak-value standard deviation is 0.05 eV. The deviation for elemental atomic concentration is associated with 0.5%. 

Surface modification with the fs laser favors the formation of Ti (IV) oxides, according to the peak fitting results (Figure 8). 

Previous studies have demonstrated that the antibacterial effect of materials is influenced by the surface contact angle [23,43]. It is more challenging for microorganisms, such as bacteria, to adhere to the surface of a hydrophobic material [43]. Wettability properties (shown in Table 2) changed from hydrophilic (non-irradiated) to more hydrophobic surfaces after irradiation with a fs laser. The changes in wetting properties are associated with both the surface relief of each sample before and after irradiation and the considerable contribution of changes in chemical composition, which favor Ti^4+^ oxide formation after laser irradiation. It can be noted that these changes to more hydrophobic surfaces have resulted in increased antibacterial activity.

The best antibacterial results were obtained for titanium plate samples irradiated with hatch spacings of 12 μm against *S. aureus* and 16 μm against *E. coli*. (Figure 9). Overall, morphological and surface chemical composition changes of the titanium surface induced by fs laser radiation led to enhanced antibacterial properties against *S. aureus* by more than 99% (Figure 10) and *E. coli* by more than 80% in comparison with the non-irradiated titanium plate sample surface. The difference in results could be explained by the differences in morphology [44] and cell wall structure [24] of certain bacteria. Spherical *S. aureus* cells have a diameter of approximately 0.5 µm and a thick peptidoglycan layer in their cell wall, while rod-shaped *E. coli* cells have a diameter of approximately 0.5 µm and a length of approximately 2 µm. Regarding the cell wall structure, *E. coli* have a thinner peptidoglycan layer and an outer membrane containing lipopolysaccharides. The superior results against *S. aureus* colonization might be explained by the lesser contact with the irradiated surface due to the spherical shape of the bacteria. 

Experiments have revealed that the antimicrobial activity of treated surfaces is derived from a synergistic interplay of various surface properties, including wettability, chemical composition, roughness, distance between LIPSS, and size of surface features.

The viral titer of the obtained, recovered, and twice-propagated Escherichia coli bacteriophage T4 stock (ATCC 11303-B4) was 2.5 × 10^8^ PFU/mL. Increasing the volume of acquired phage stock resulted in a correspondingly lower phage concentration—that is, the output phage titer for further experiments was 2.5 × 10^7^ PFU/mL. 

The phage stability was maintained at all time points (0 h, 6 h, 12 h, 24 h, and 36 h) assessed by plaque assay; no log loss in the phage titer was observed (see Figure 11 and Figure 12). Furthermore, all experiment samples, namely, phage-only suspensions in TSB, Tref, and T16 after incubation for 36 h, exhibited slightly increased viral titers: 3.5 × 10^7^ PFU/mL, 3.6 × 10^7^ PFU/mL, and 3.8 × 10^7^ PFU/mL, respectively. In addition, the viral titer rose more rapidly in all experiment samples after 12 h of incubation.

## 4. Conclusions

LIPSS were formed on the titanium plate sample surface with a periodicity in the range of about 520–740 nm and a height in the range of 150–250 nm. Changes in the surface chemical composition took place, including the formation of Ti (IV) oxides after irradiation with a fs laser. The changes in the wetting properties of more hydrophobic surfaces have led to increased antibacterial activity. The experimental results revealed that morphological and surface chemical composition changes of the titanium surface induced by fs laser radiation led to enhanced antibacterial properties against *S. aureus* by more than 99% and *E. coli* by more than 80% in comparison with the non-irradiated titanium plate sample surface. 

Regarding the interaction between unmodified and modified Ti plates and bacteriophages, the key finding in our study was that Ti plates do not possess virucidal activity against bacteriophages before and after irradiation. There are potential limitations to using phages as prototypes to evaluate the antiviral activity of Ti plates, as they are prokaryotic viruses. However, no loss of phage titer was observed in the presence of Ti plates, and phage stability was preserved throughout the experiments. 

Considering the antibiofilm effect of phages on biofilm viability, which plays a crucial role in the pathogenesis of implant-associated infections, the results of our study indicate a potential application of Ti implant devices concurrent with phage-based films, preventing possible loss of the infectivity titer of phages. Moreover, the findings on maintaining the stability of phages may be a starting point for future research to investigate whether Ti plates even potentiate the lytic activity of phages. In that event, phage-based antimicrobial coatings and Ti plates in biomedical implants would ensure a potent anti-infective approach in therapeutic and preventive interventions for implant-associated infections. Rigorous future investigations are vital to address the antibacterial interplay between phages and Ti and assess whether they interact in an additive, synergistic, or antagonistic manner. 

## Figures and Tables

**Figure 1 nanomaterials-13-02032-f001:**
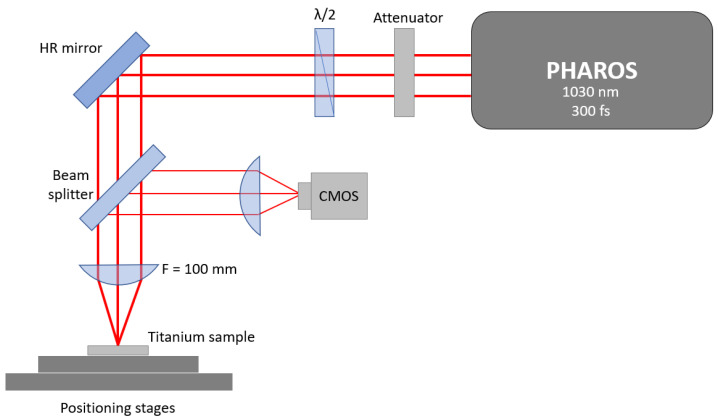
Scheme of the experimental laser setup used for irradiation of titanium plate samples for the formation of LIPSS: CMOS—optical camera; HR—high reflection mirror.

**Figure 2 nanomaterials-13-02032-f002:**
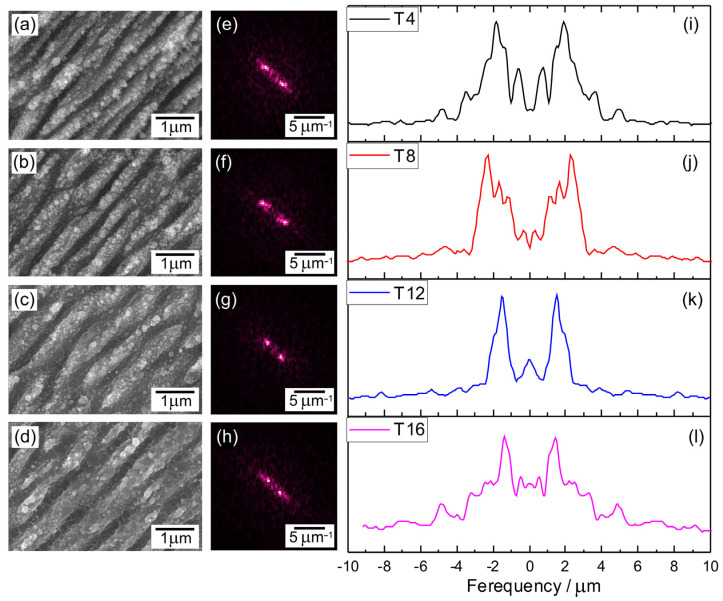
Graphical representation of the LIPSS periodicity measurements procedure: FESEM images of fs laser irradiated titanium plate surfaces with hatch spacings of 4 (**a**), 8 (**b**), 12 (**c**), and 16 (**d**) µm with corresponding 2D-FFT images (**e**–**h**) of the SEM image and profile (**i**–**l**) from 2D-FFT images with peaks representing spatial frequencies.

**Figure 3 nanomaterials-13-02032-f003:**
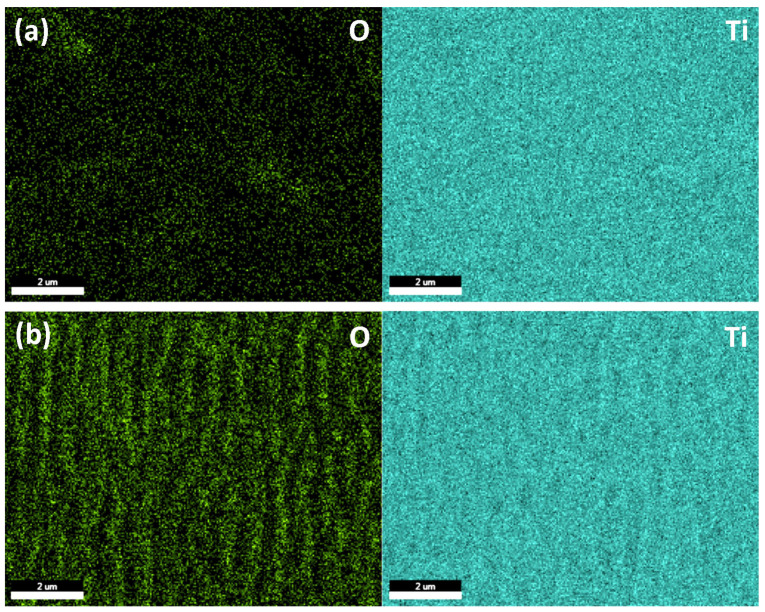
EDS mapping of non-irradiated (**a**) and fs laser irradiated (**b**) titanium surfaces. The green color in the elemental maps corresponds to oxygen (O) and blue to titanium (Ti).

**Figure 4 nanomaterials-13-02032-f004:**
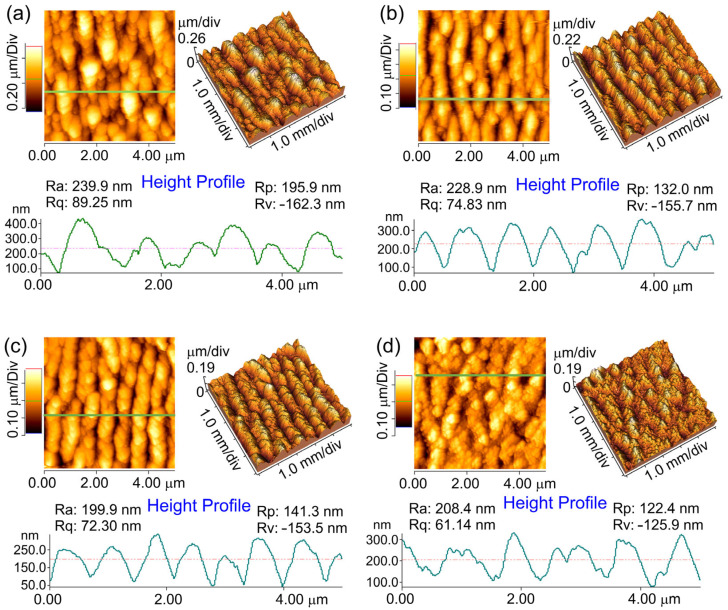
AFM images and line profiles of fs laser irradiated titanium plate surfaces with hatch spacings of (**a**) 4, (**b**) 8, (**c**) 12, and (**d**) 16 µm.

**Figure 5 nanomaterials-13-02032-f005:**
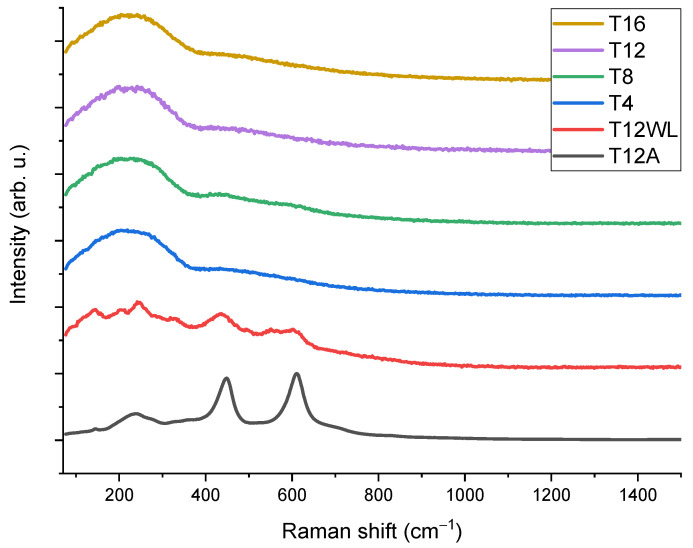
Raman spectra of titanium plate surfaces irradiated with a femtosecond laser with different laser parameters.

**Figure 6 nanomaterials-13-02032-f006:**
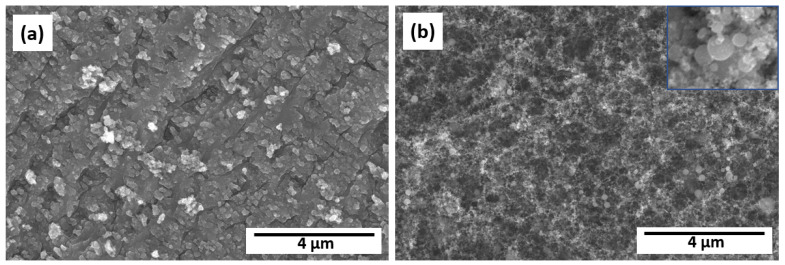
FESEM images of fs laser-irradiated titanium plate surfaces T12WL (**a**) and T12A (**b**). The inset in the top right corner of the image (**b**) shows an enlarged view of the morphology of nanoparticles with a horizontal width (HFW) of 300 nm.

**Figure 7 nanomaterials-13-02032-f007:**
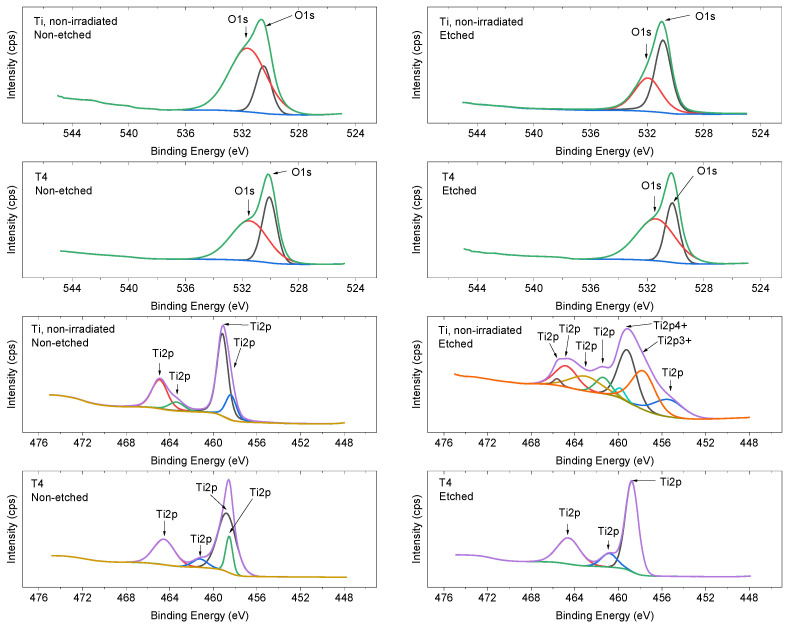
XPS spectra of titanium plates before and after irradiation with a fs laser. On the right side, the image shows the non-etched surface, while on the left side, the image shows the surface after etching using an Ar ion gun.

**Figure 8 nanomaterials-13-02032-f008:**
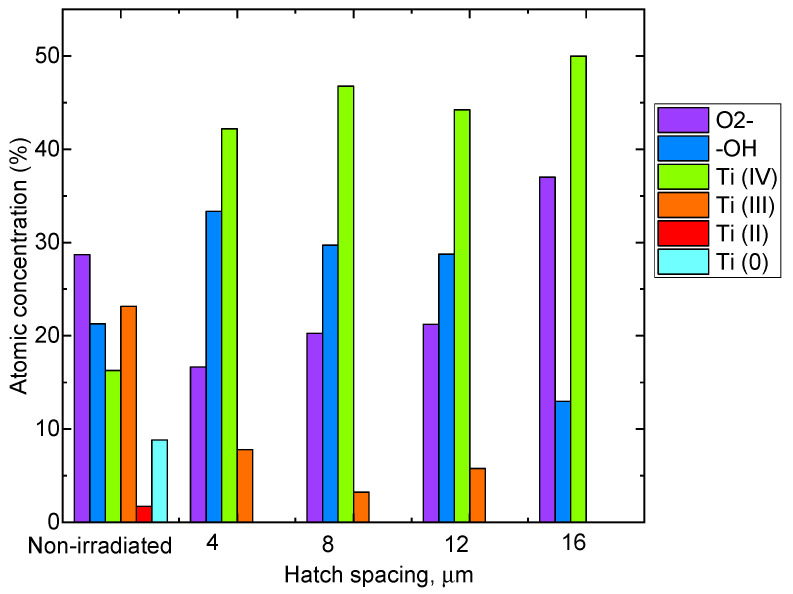
Relative atomic concentrations from XPS analysis of non-irradiated and irradiated samples with different hatch spacing.

**Figure 9 nanomaterials-13-02032-f009:**
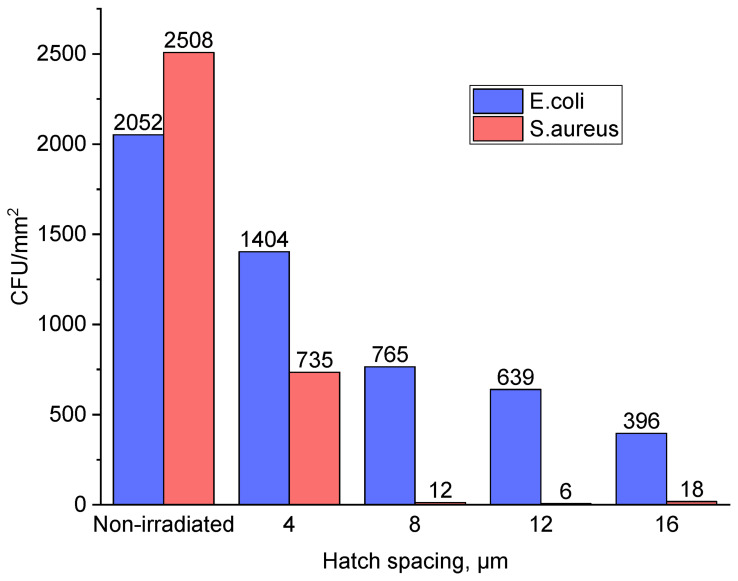
Antibacterial activity of titanium plates against both Gram-positive *Staphylococcus aureus* and Gram-negative *Escherichia coli*.

**Figure 10 nanomaterials-13-02032-f010:**
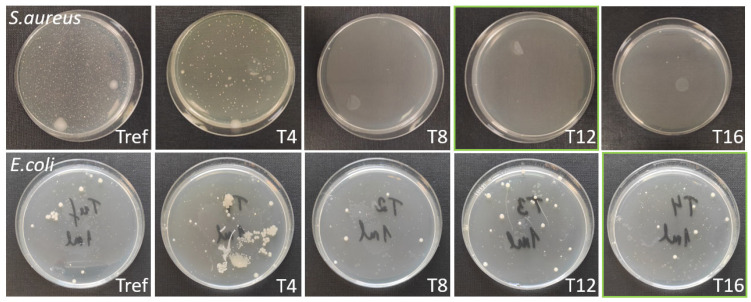
Colonization of *S. aureus* and *E. coli* on the Ti plate surface before (Tref) and after fs laser treatment using different hatch spacing. The diameter of the Petri dishes is 85 mm. The green squares represent the best results obtained in the cases of *S. aureus* and *E. coli*, respectively.

**Figure 11 nanomaterials-13-02032-f011:**
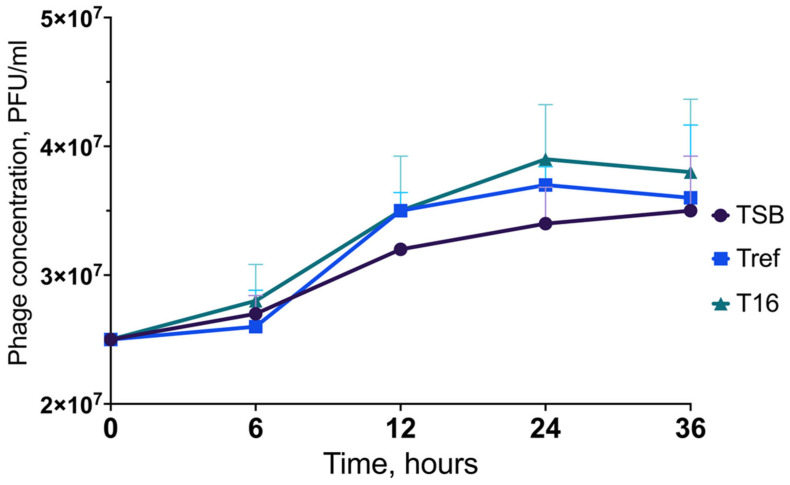
Preserved stability of *Escherichia coli* bacteriophage T4 (ATCC 11303-B4) exposed to a T16 titanium plate over a 36 h period. TSB—phage-only suspension in TSB; Tref—phage suspension with Tref titanium plate; T16—phage suspension with T16 titanium plate. The statistical tests resulted in *p*-values greater than 0.05, considering the differences in the results obtained as not statistically significant.

**Figure 12 nanomaterials-13-02032-f012:**

Visual plaque assay results of *Escherichia coli* bacteriophage T4 (ATCC 11303-B4) over a 36 h period. (**A**) Plaques of a 10^4^ dilution of an original phage stock at 0 h. (**B**) Plaques of a 10^4^ dilution of a phage stock when exposed to a T16 titanium plate for 6 h, (**C**) 12 h, (**D**) 24 h, and (**E**) 36 h. The diameter of the Petri dishes is 85 mm.

**Table 1 nanomaterials-13-02032-t001:** Laser process parameters.

No	Laser Fluence, mJ/cm^2^	Spot Size, μm	Laser Rep. Rate, kHz	Scanning Speed, mm/s	Hatch, μm
T4	63	80	600	1	4
T8	63	80	600	1	8
T12	63	80	600	1	12
T16	63	80	600	1	16
T12A	254	40	600	2	12
T12WL	106	40	600	2	12

**Table 2 nanomaterials-13-02032-t002:** Wetting properties depending on hatch spacing.

Sample	Tref	4 μm	8 μm	12 μm	16 μm
Contact angle	79.5	136.5	112.0	98.7	104.5
(°)	±2.3	±2.4	±1.5	±3.0	±2.9

## Data Availability

The data that support the findings of this study are available upon reasonable request from the authors.

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
