# Peer review of "Effect of Femtosecond Laser-Irradiated Titanium Plates on Enhanced Antibacterial Activity and Preservation of Bacteriophage Stability"

_nanomaterials, 2023, doi:10.3390/nano13142032_

Round 1

Reviewer 1 Report

In this manuscript of Grase et al. demonstrate interesting results on femtosecond ablation of titanium and its effect on antibacterial activity. The materials were well characterized: SEM, XPS, AFM etc. and the proof-of-concept demonstrated (S. aureus and E. coli vs bacteriophage T4).  Some points should be mentioned, prior to acceptance of this work for publication.

1. In Introduction Authors should mention other application of fs laser, e.g. making SERS plaforms: Szymborski et al.

https://www.sciencedirect.com/science/article/pii/S2238785421003100

2. How the SEM samples were prepared? any additional surface of carbon, gold? if not, it should be also mentioned.

3. What was the source of E. coli and S. aureus? they were bought, received from another lab?

4. Figure 5: please put description of T16, T12 etc in the Figure.

5. Please improve quality of Figure 9

6. Please add scale in Figure 10 and Figure 12

7. Authors have to elaborate more on the results in the Conclusions. What are potential applications of this knowledge presented in the manuscript? does the method offers something new?

English language is fine - please re-read the whole manuscript before re-submission

Author Response

Dear Reviewer,

We are grateful for our manuscript revision and your valuable comments and suggestions.

Kind regards,

Reviewer 2 Report

I have read this paper carefully. This paper has described the effect of femtosecond laser-irradiated titanium plates on enhanced antibacterial activity and preservation of bacteriophage stability. I have some strong opinions to share with the author.

1.      The period and height of the LIPSS structure are mentioned in the abstract of the article, which is inconsistent with what is mentioned in the results and discussion. Also, it is inconsistent with what is mentioned in the conclusion.

2.      The wavelength of the femtosecond laser selected in the article is 1030 nm and the half-wavelength is 515 nm. The period of the LIPSS structure experimentally measured by the authors is 300-850 nm, which is contradictory to the description that the period of the LIPSS structure is less than half of the laser wavelength. The period of LIPSS (HSFL) is less than half of the laser wavelength (), which is not mentioned in the text.

3.      The images in the article are not clear and it is necessary to adjust the resolution of the images.

4.      The material chosen is the commercial material Ti. The article mentions that T16, T12, T8, and T4, appear as amorphous semiconductor phases, and T12WL appears as a transition between amorphous and crystalline phases. Please describe the specific form of the transition in detail.

5.      The nanoparticle diagram depicted in Figure 6(b) in the article does not have a scale and needs to be labeled.

6.      Wetting tests are not convincing and require more detailed data such as photographic evidence.

7.      The text describes the LIPSS results for S. aureus and E.coli inhibition, but Figure 10 only presents the inhibition effect for S. aureus.

1.      Check grammar and English spelling in the text.

Author Response

(The authors gave the same response as above.)

Reviewer 3 Report

The manuscript presents laser treatment of Ti metal surfaces which results in the formation of laser induced periodic surface structures. The authors have treated the surface with different laser irradiation conditions 9regarding hatch period) and have used the resulting modified surfaces to demonstrate the antimicrobial function of the surface.

The authors have investigated the nature of the resulting structures using various spectroscopic and microscopic methods, which represents a large amount of data.

Having read the manuscript, I have come to the conclusion that it does not merit publication in nanomaterials for the following reasons:

Lack of novelty; I am unsure where the novelty of this work is located, whether it is associated with the patterning, or the antibacterial function. It cannot be the former as there is prior art in the literature on the LIPSS patterning.

Lack of Benchmarking; To correctly evaluate the antibacterial action of the modified surfaces it is necessary to benchmark the results by comparison to other reports in the literature.

Lack of focus; this paper tries to address two tasks i) investigate the laser based pattering and ii) study the antimicrobial behaviour. There is no clear centre of mass and none of the two tasks is addressed appropriately.

In spite of the large amount of data that are presented in the manuscript, it seems to me that there is no critical and in depth analysis of the results.

Here is a list of more specific comments

·      LIPSS on Ti metal surfaces have been observed before, the authors cite some of the older work. I could also add “Paulina Segovia, Abraham Wong, Ricardo Santillan, Marco Camacho-Lopez, and Santiago Camacho-Lopez, "Multi-phase titanium oxide LIPSS formation under fs laser irradiation on titanium thin films in ambient air," Opt. Mater. Express 11, 2892-2906 (2021)”

·      This study is inconclusive regarding the origin of the antimicrobial activity of the treated surfaces. There does not to be a clear evidence of what is the origin of the observed behaviour.

·      The antibacterial function is corelated with the hatch distance only. But there is lack of systematic investigation that would lead to correlation with other parameters such as the chemical changes of the surface as they are revealed from XPS, Raman etc.  

·      The authors claim changes of the wetting properties of the surface but they do not provide any contact angle measurements.

·      There is seem to be some confusion about the period of the LIPSS structures that have been achieved. In the text the period is indicated to be in the range of 630-790 nm, but in the conclusions it is mentioned that the period are in the range of 300-850 nm. How was the period measured? Did the authors use Fourier analysis?

·      What is the polarization of the beam that was used in the experiments

·      These period range that is quoted in the manuscript does not qualify for HSFL. HSFL periods correspond to ~ lamda/10. The period of the LIPSS can vary depending on the dielectric properties of the material that is being treated and also by the angle of irradiation.

Author Response

(The authors gave the same response as above.)

Round 2

Reviewer 3 Report

Having read the revised manuscript and the author’s response, I still have questions about the quality of this report.

Undoubtedly, this work represents a large amount of experimental work, however the analysis of the results in the revised manuscript has not improved significantly. There is still a question about what is the combination of factors that result in the antibacterial function. There are speculations sprinkled in the discussions and the conclusions, but a serious in depth discussion is still missing.

For example, hydrophilicity alone does not correlate with the observed antibacterial function as we can see the sample with the strongest antibacterial behaviours does not exhibit the largest contact angle.

It is reported that there are chemical changes on the surface, evidenced by Raman and XPS, but how do they contribute to the antibacterial function of the treated surface? Is there any evidence in the literature for that?

Regarding LIPSS, it is important to know the direction of laser polarisation with respect to the direction of the periodic features, in order to compare with the literature. Also, the FFTs should be displayed to provide an idea of the uniformity of the periodic patterns.

It is still my opinion that this article, despite the wealth of data, does not go in depth and therefore its overall contribution to the subject is rather poor.

Perhaps the authors could provide a paragraph with a clear discussion, where ALL the results are considered collectively under the light of existing literature and point concretely at the contributors to the antibacterial behaviour.

Author Response

(The authors gave the same response as above.)

Round 3

Reviewer 3 Report

The manuscript has now been sufficiently improved to qualify for publication